# What Are the Current Directions in the Local Marketplaces Fiscalization? The Online Media Content Analysis

**Stefan Denda** [1,*], **Marko D. Petrović** [1,2,*], **Zlata Vuksanović-Macura** [1], **Milan M. Radovanović** [1] **and Edna Ely-Ledesma** [3]

1   Geographical Institute "Jovan Cvijić" SASA, 11000 Belgrade, Serbia; z.macura@gi.sanu.ac.rs (Z.V.-M.); m.radovanovic@gi.sanu.ac.rs (M.M.R.)
2   Department of Regional Economics and Geography, Faculty of Economics, Peoples' Friendship University of Russia (RUDN University), Moscow 117198, Russia
3   Department of Planning and Landscape Architecture, University of Wisconsin-Madison, Madison, WI 53706, USA; eledesma@wisc.edu
*   Correspondence: s.denda@gi.sanu.ac.rs (S.D.); m.petrovic@gi.sanu.ac.rs (M.D.P.)

**Abstract:** Local markets have been a special setting throughout human history. Apart from their important social role, they had immeasurable economic importance as primary forms of exchange of goods (trade). Nonetheless, they experienced numerous transformational changes that affected their functioning. Like other countries, Serbia has a long tradition of market activity. However, several novelties have been introduced in recent years. Among many, the process of e-fiscalization is the main issue. Therefore, the focus of our research is based on a qualitative analysis of online media content (news and comments) related to the fiscalization of market activity. The attitudes of different categories of participants (state authorities, vendors, and customers) were analyzed. LIGRE open-access software was used for this purpose. The results of the analysis showed conflicting parties. Legislators emphasize the exclusive positive effects, while vendors point to the negative side of fiscalization. As a third party, customers (service users) have an undefined attitude in relation to fiscalization (pros/cons/neutral). There is an agreement to introduce market activity into legal flows. However, the key prerequisite is the prior resolution of a number of problems (working conditions, business costs, market monopoly, etc.).

**Keywords:** marketplaces; fiscalization process; online media content; public opinion; qualitative analysis; open coding; Serbia





## 1. Introduction

Throughout human history, markets (*local marketplaces, bazaars*) have been the main gathering places for vendors and customers. Located in attractive places, in the centers of urban and rural settlements, markets have historically enabled the exchange of various goods and services. They represent a unique form of the lives and work of the populations they serve, and others who participate in this process of exchange [1]. This provides markets with the outstanding social role of connecting people since they enable communication between numerous actors from the local community. By organizing events and similar activities, in addition to their primary role of economic transaction, they have become an inevitable part of the social life of a community. Their role goes beyond the "vendor–customer" context with their contribution to the quality of life of the local community members. They create a new sense of a local living culture, becoming an indispensable element of the area's intangible cultural heritage [2]. Simply, they have become an integral part of society through their improvement of public places and their introduction of novelties, in addition to their role in fostering businesses [3,4].

Markets take on a variety of forms, such as open and closed (and covered), permanent, periodical, and mobile, as well as other forms (e.g., wholesale markets) [5]. Based on the

type of products, most of them are miscellaneous markets (agricultural food products), goods or artisanal markets, wholesale markets (wholesale fruits and vegetables), cattle markets, and used-car markets. In terms of organization, technological advancements and other circumstances (economic, ecological, social, legal, etc.) have also led to their transformation over time. Nevertheless, the role of directly connecting consumers and vendors is what differentiates markets from other commercial shops.

In sociological terms, markets are the best indicator of the socioeconomic position of the population and the development level of economic activities in a country [6]. The variety of products, direct communication, and market dynamics make this way of trading unique in comparison with other forms of consumer exchange. However, for more than half a century, a key challenge faced by traditional markets has lain in the new forms of supplying goods, such as through supermarkets. In particular, marketplaces are now marginalized as a result of internet shopping, city redevelopment, and the globalization of retail. Additionally, they are becoming places of urban conflict. Therefore, in modern times, marketplaces have a triple role as (1) frontlines in the gentrification, eviction, and dispossession processes; (2) forums for protest, mobilization, and discussion about the city and public space; and (3) environments for developing counter- and alternative modes of production and consumption [7]. The COVID-19 crisis represented a new challenge. In addition to their temporary closure, a number of different measures were applied to the operation of markets (improving hygiene conditions, social distancing among vendors, etc.) [8,9]. Apart from the highlighted socio-economic challenges expressed through rising prices, the COVID-19 crisis affected the everyday geographies of public space [8,10]. In general, the food supply chain has been improved. Local and regional food systems have been established in a number of countries (for example, "Small producers" in Serbia) [11,12]. But besides all the unsolved questions and challenges of various natures (legal, economic, political, social, etc.), markets have continued to survive in our modern market environment. This has been accredited to their vitality and adjustability [13]. That is why we often use the common Serbian phrase that "a market has its soul because people make a market".

The informal sector has a large share in the economies of many countries. It exerts a significant influence on the sustainability of development. The aim of the state (tax administration) is to lessen the amount of informality in commerce and other economic endeavors and direct the country towards a formal economy [14]. The goal of any tax authority is to eliminate all forms of the informal economy (grey market). Consequently, among the most critical issues, the regulation of fiscal policy stands out. However, there is still no consensus regarding the fiscal burden of both traditional and conventional business models [15,16]. Although it is not part of the "classic" informal sector, marketplace activity is not included in the tax system in a large part of the world. Their exclusion, in a certain way, puts them in a superior position in relation to other actors in the trade sector. Aside from the advantages for the supply side (vendors/traders), certain benefits are also present on the demand side (customers). They are, first and foremost, participants who are looking to find a more convenient bargain (i.e., lower price or greater value for money) [17]. Apart from being an economic challenge, the topic of electronic fiscalization also has political implications. As stated by Gottlieb [18], the inconsistency of fiscal policy leads to mistrust in state administration and, consequently, to the impossibility of programmatic coordination between stakeholders.

Similar to other countries, Serbia has a tradition of "market business" through selling agricultural and other products. The provisions of the Food Safety Law [19] enable markets to process and place goods. Thus, urban markets directly "keep" the value of the production and income in the local environment and indirectly contribute to preserving villages and rural areas, particularly those located in undeveloped and devastated regions of Serbia. In the early 2000s, the economic landscape of Serbian markets began to be threatened. Here, the question of fiscalization was raised for all economic activities, including the market sector [20]. Although this taxation initiative failed, in 2020, the idea of the fiscalization of

a number of activities (including communal activities) was reintroduced, which caused a division in public opinion.

The introduction of fiscalization represents an important social and economic issue in the context of the modern market economy. Several reasons have influenced such a situation. First of all, the regulation of market activity through the introduction of VAT (value-added tax) represents the continuation of activities that started in the 2000s. The involvement of a large number of stakeholders (state authorities, vendors, customers, and professional associations) with different interests caused a sharp polarization in society. Namely, the primary intention of the state is to integrate this activity into legal economic flows. However, although there is an agreement in principle to resolve this issue, the proposed steps for the other side (vendors) are not acceptable. In such a situation, customers (the third party) put themselves first, respecting both opinions (administration and vendors). As a fourth actor, professional associations strive to balance the situation between the "opposing" parties in which both would be satisfied to a certain extent. Considering all the above, the fiscalization process is becoming an increasingly important political topic in an emerging market such as Serbia.

Therefore, this study concentrates on the historical evolution, the process of electronic fiscalization of market activities, from law proposal to realization, and the economic importance of markets. The main goal is to determine the distinctions and parallels between the social and economic outlooks of the markets using theoretical presumptions and qualitative content analysis at an important historical moment for local markets in Serbia. Given the division in public opinion, we aim to present new measures proposed by the government and, using qualitative content analysis, evaluate the public reactions to the e-fiscalization of market activities. Moreover, more research is needed to deal with the topic of e-fiscalization in local market businesses. Given the absence of prior research on this topic in Serbia and the Balkans [21,22], this study may serve as a starting point for more in-depth research. Furthermore, the study is relevant since it is the first empirical investigation on the cohesion between financial and social perspectives and the function of marketplaces in the process of modernizing the Serbian economy and society.

## 2. Background of Markets in Serbia

### 2.1. Historical and Social Significance of Markets in Serbia

The first forms of markets in Serbia were connected with "fairs", especially since the 18th century. People usually gathered there during the weekends and religious holidays. Prior to the appearance of modern retail chains, these historic markets were the only places to exchange affordable (competitive), fresh, and health-safe food. This historic mode of market exchange continued for much longer in Serbia than in other locations due to the economic backwardness of the Balkans, as well as the incomplete transition process, which started at the end of the 1990s [6].

Since markets in Serbia have both communal and market features, their business is regulated by the Law on Communal Activities and the Law on Trade. According to the Law on Communal Activities [23], markets are organized to provide the population with fresh food and other food and non-food products. As an integral part of the communal infrastructure, they belong to the activities of public interest. Accordingly, the management of markets includes communal equipping, maintenance of the ancillary facilities (business premises, booths, and stalls in the open air), and stall renting, as well as the organization of trade activities for selling and buying agricultural goods and other products. In accordance with the given Law, every local self-government adopts the Decisions on Markets [24]. They define the activity, maintenance (market rules), equipping, use of facilities, equipment, and space, as well as the supervision of the implementation of the articles of these decisions (including penalty clauses for provision breaching). The most important internal act for communal enterprises is the market rules [25], which regulate the following aspects: trade of goods and provision of services in the trade of goods on the market; market business premises, facilities, and equipment, as well as their conditions and the ways of their use;

hygienic/sanitary and technical conditions (hygiene maintenance); work hours and rights and obligations of the participants in the trade within the markets.

Additionally, Article 22 of the Law on Trade [26], refers to markets as special market institutions. They organize retail sales through the arrangement, maintenance, and issuing of specialized spaces for conducting market sales. According to the Law, market sales of goods are carried out on stalls, in booths, and in specialized sales facilities. Except for the sales of agricultural food items, it comprises the sales of handicraft and artisanal products, as well as the provision of accompanying services. Based on the Law on Agriculture and Rural Development [27], agricultural products are marked as primary products and products whose first level processing involves agricultural production. Markets play a role in facilitating the placement of products from small producers.

Nowadays, around 160 legal entities, such as public companies, conduct market business in Serbia. They manage about 410 markets, which are comprised of agricultural producers and resellers. They are mainly (over 90%) managed by (local) public companies in the jurisdiction of the relevant local self-government units. The majority of markets are so-called miscellaneous and green markets (61%). Out of a total number of 70,000 selling places in Serbia, 74.3% are covered and non-covered market stalls, 17% are designated selling places, 8% are small shops, 5% are specialized stalls with refrigerated showcases, and 2% are seasonal booths for fruit and vegetables [28]. The most significant legal entity which conducts market activity is the "Belgrade markets" Public Utility Company (*JKP* "*Gradske pijace*") in the capital, the City of Belgrade. It manages 30 markets, out of which 28 are green markets with 11,531 selling places (16.5% of the total number) [29]. Together with two public companies from two other large national city centers, "Tržnica" from Novi Sad and Niš, it makes up over 28% (about 20,000 selling places) of the total selling capacity in Serbia. At the same time, about 7% of "small" markets are located in settlements with fewer than 7000 inhabitants, which the municipal offices manage [16]. These markets provide a daily offering of food products (fruit and vegetables, milk and dairy products, meat and meat products), as well as artisanal products and consumer goods (so-called "flea markets"). The exceptional qualitative contribution of markets is reflected in the distribution of the products of small food enterprises. They include ready-made food products of agricultural origin (fruit and vegetable preserves, dairy products, preserved meat products prepared according to the traditional recipe, honey and honey products, flour, and the accompanying products, teas, etc.).

With the aim of improving business, a professional "Business Association of Serbian Markets" was founded in Novi Sad in 2003. Twenty years later (in 2023), it is comprised of twenty-six permanent and six associated members from all parts of the country, which makes it around 70% of all of the market potential [30]. In the context of reconstruction and transformation, the companies which do their business in markets are facing the possibility of a change in the ownership structure. Thus, "Business Association of Serbian Markets" believes that markets should remain in the ownership of the founders (towns/cities and municipalities). Nevertheless, in accordance with the Law on Communal Activities [23] and the Law on Public-Private Partnership and Concessions [31], the privatization and public–private partnership of part of the market activity should be enabled.

In recent years, distribution channels for fresh food and products from small food enterprises have widened. At the state level, to address the limited distribution of food and other goods caused by the COVID-19 pandemic [9], at the beginning of April 2020 (April 10), the Ministry of Agriculture, Forestry, and Water Management launched a portal entitled "Electronic (e-)market of Serbia". A simple registration enabled all the users to buy fruit and vegetables, meat, milk and cheese, eggs, honey, and other products and to have them delivered by the producers or courier services [32]. By January 2024, this project comprised 1389 producers from 13 Serbian regions, which offer 16 different types of products [33].

*2.2. Economic Importance of Markets in Serbia*

Historically, local marketplaces have had a significant economic role. In Serbia, around 80,000 people are engaged in sales at marketplaces every day. At the same time, the data on the turnover value paints a challenging picture. The Statistical Office of the Republic of Serbia publishes monthly reports on the turnover value of agricultural products in green markets. This report provides data about the direct sales of products performed by agricultural holdings (excluding sales through retail trade shops and individual trade shops). These reports show that the turnover value for those buying up agricultural products or participating in selling at green markets in Serbia has been, on average, 20–35% of the total for the past several years [34].

The total turnover value of agricultural products at this type of market in Serbia (2018–2023) was about 2.37 billion USD (Table 1). From 2021, a stable growth in turnover value has been recorded after the market collapse caused by the COVID-19 crisis. At the level of different regions in the country, in five years (2018–2022), the Belgrade region had the largest average share (46.0%), followed by Šumadija and West Serbia (20.8%), Vojvodina (17.7%), and finally, South and East Serbia (15.5%) [34]. Similar trends continued during 2023. The share of the Belgrade region exceeded half of the total turnover (51.4%). The only exception was the region of South and East Serbia (11.1%). The reasons for the decrease can be seen from several aspects (the least developed regional unit, bad demographic indicators, low purchasing power, etc.).

**Table 1.** Total turnover value of agricultural products on green markets in Serbian regions (in mln. USD) *.

| Name of the Country/Regions | 2018 | 2019 | 2020 | 2021 | 2022 | 2023 |
|---|---|---|---|---|---|---|
| Republic of Serbia | 342.97 | 374.36 | 350.91 | 367.41 | 450.32 | 484.06 |
| Belgrade | 154.21 | 161.46 | 161.27 | 177.02 | 215.33 | 248.73 |
| Vojvodina | 60.34 | 63.22 | 61.98 | 70.00 | 77.54 | 86.88 |
| Šumadija and West Serbia | 79.86 | 87.03 | 70.02 | 64.32 | 90.07 | 94.49 |
| South and East Serbia | 48.56 | 62.65 | 57.64 | 56.07 | 67.38 | 53.95 |
| Kosovo and Metohija ** | − | − | − | − | − | − |

* Exchange rate: 1 US dollar (USD) = 109.74 Serbian dinnar (RSD). ** The data relating to the Kosovo and Metohija region are not included due to the impossibility of collecting. All references to Kosovo and Metohija in this paper should be understood to be in the context of United Nations Security Council Resolution No. 1244 (1999). Source: [34].

The most significant drop in the turnover value of markets in Serbia, expressed in current prices, compared to the previous year, was in 2020 (6.3%). The regions of Šumadija and West Serbia can be pointed out with a decline of 19.5% each, while the Belgrade region did not record significant changes in agricultural turnover value (Table 2). However, the largest growth in turnover in current prices at the national level was registered in 2022 (22.6%). The Šumadija and West Serbia regions should also be pointed out individually (40.0%) [34]. On the other hand, data from 2023 indicate the stability of market trends in the regions of Belgrade and Vojvodina. At the same time, the value expressed in current prices has significantly decreased in the regions of South and East Serbia (40.0%), as well as Šumadija and West Serbia (35.0%) [34]. These indicators should be observed conditionally because market changes must be carefully monitored over a longer interval.

**Table 2.** Total turnover value of agricultural products on green markets in Serbian regions (indices, current prices).

| Name of the Country/Regions | 2018/2017 | 2019/2018 | 2020/2019 | 2021/2020 | 2022/2021 | 2023/2022 |
|---|---|---|---|---|---|---|
| Republic of Serbia | 106.1 | 109.2 | 93.7 | 104.7 | 122.6 | 107.5 |
| Belgrade | 107.5 | 104.7 | 99.9 | 109.8 | 121.6 | 115.5 |
| Vojvodina | 102.7 | 104.8 | 98.0 | 113.0 | 110.8 | 112.0 |
| Šumadija and West Serbia | 108.5 | 109.0 | 80.5 | 91.9 | 140.0 | 104.9 |
| South and East Serbia | 100.1 | 129.0 | 92.0 | 97.3 | 120.2 | 80.1 |
| Kosovo and Metohija ** | — | — | — | — | — | — |

** The data are not included due to the impossibility of collecting. Source: [34].

Consequently, the capacity and use of markets was smaller by 12% (about 56% in 2021), and the rent collection dropped by about 5%. This caused the income of communal companies that deal with market management to fall by 25.5%. It should be noted that COVID-19 significantly affected the business of markets, taking into account the fact that it urged the further strengthening of other forms of trade (primarily supermarkets) [34]. Also, today, there are a lot of foreign and domestic retail chains which conduct business in Serbia and have megamarkets throughout the country [35]. Additionally, within the Association of Domestic Retail Chains (founded in 2011), there are 14 trading companies with over 7500 facilities and 6000 employees [36]. Their presence (monopoly position) has a growing negative impact on green market activity.

*2.3. Introducing Regulations for the Electronic Fiscalization of Market Activities*

Recently, market activity has been in a phase characterized by rapid changes. Besides the more intensive competition of other forms of trade and the changes in the consumers' needs and habits [37], the crucial elements relate to the changing legislation in the field of fiscal policy. However, apart from numerous changes in previous years, the market activity remained outside of this system. A study on the scope of the gray (shadow, underground) economy in 2017 showed that it was necessary to introduce fiscal cash registers to all economic activities and for all of the business subjects that provide services [38]. In the follow-up research on citizens' attitudes toward the gray economy, the majority of participants (53%) thought that issuing fiscal receipts was necessary (including market stalls). Almost 72% emphasized the need for contactless payment (use of cards) in comparison with the traditional payment method (cash) [39].

Thus, around the middle of 2021, a discussion was begun on the implementation of the Law on Fiscalization, which should have started in January 2022. The participants of the debate were market managers, representatives of the Ministry of Finance and Tax Administration, trade associations within the Serbian Chamber of Commerce, and the Business Association "Pijace Srbije". It was pointed out that the service users (author's note: vendors) should make all the information available and give instructions related to the implementation of the new regulations. Such a process was completed in a short time, bearing in mind that it has been estimated that about 10,000 entrepreneurs work in markets. Of that number, about 3000 keep a record of their turnover using a device, while the rest were not obliged to. Besides the majority of entrepreneurs who work in goods and miscellaneous markets, some merchants deal with retail sales of food, beverages, and tobacco products on their stalls [30].

After several sessions of discussions, the adopted amendments to the Law on Fiscalization [40] defined the scope of (retail trade of goods and services and the accepted advance payment for the retail trade) and the tax that should be paid on income from independent activities. The Law also defined a transition period (1 November 2021–30 April 2022) for the compliance of businesses and for those persons that did not keep records of each individual sale or that had not been not obliged to keep a record of their turnover using a fiscal cash register. These amendments were considered "historic or (epochal)" since the list of subjects obliged to issue fiscal receipts was expanded. Besides those doing

business through postal service, the internet, and so-called e-commerce, the obligation to issue fiscal receipts was also extended to the market vendors. The original solution included agricultural producers (agricultural households) and those merchants who resell products. The Serbian Government later issued an adequate Act [41], which removed the obligation to keep records on retail sales and to acquire advance payments using an electronic fiscal device.

As a flexible system, e-fiscalization is described as a much simpler and more convenient method for all actors. According to the opinion of the Ministry of Finance, the most important advantage is reflected in a more effective fight against the gray economy, as well as in an easier way of conducting business (less administrative work), reduced costs, and the creation of a better business environment. It also enables the monitoring of business facilities in real-time [42]. By using the new hardware and software solutions, it is possible to issue fiscal receipts with QR codes, which enable customers to review the validity of their receipts [43]. To facilitate implementation, in September 2021, the Government adopted the Act with the aim to provide financial support to fiscal payers. In that respect, the existing and new payers are entitled to use the assets of 107 USD per registered business facility and per each registered active fiscalized cash register [44]. In order to inform the business people and citizens, a campaign entitled "Be e-fiscalized" (*Budi eFiskalizovan*) was launched with the slogan "Short reckonings make long friends". Also, for the exact details, a dedicated website was created with the title "Be e-fiscalized" [45].

According to the opinion of tax experts, the introduction of innovations, in this case fiscalization, is aimed at having more effective control over business subjects and better tax collection. It will contribute to the suppression of the gray (underground) economy and the creation of more equal conditions for the actors in the market, taking into consideration that there are a lot of economic activities that are not in the public budget system. However, the country has changed its fiscalization model many times in recent periods. Experts pointed out the lack of cost–benefit analysis as the main reason [46]. There is a reasonable risk that the introduction of fiscalization into activities with a small turnout (author's note: market sales) will have an opposite effect and cause the growth of a gray economy.

Naturally, there was a risk that the Law could affect the business of market management throughout Serbia, leading to a reduction in income from the rents of stalls and business facilities. Due to that risk, certain communal enterprises considered the possibility of correcting their prices. Thus, they would keep not only resellers but also the customers who increasingly turn to the more modern shopping centers equipped with better infrastructure. Also, the most "illogical" thing was the exemption of lawyers and taxi drivers, while it included other service-providing activities with a flat-rate tax (e.g., e-commerce). In short, the opinion prevailed that the existing model was not "just" [47].

Due to numerous doubts, in the period 7–21 February 2022, together with the Serbian Chamber of Commerce, the Business Association "Pijace Srbije" carried out a survey on the topic of the fiscalization of markets. The results showed that more than half (54.5%) of the participants thought that there were no conditions for working with fiscal cash registers on markets. As the main reason for not applying for e-fiscalization, they mentioned the fall in turnover (38.6%) and the growth of business costs (26.9%) [48].

At the beginning of March 2022, the Ministry of Finance clarified the Act in force (at that time), according to which resellers have to have fiscal cash registers, but the agricultural producers who sell their goods on market stalls do not [41]. Based on the records, there are 3663 of them in total in Belgrade markets, while the number of resellers is 1151 (223 in the food sector). The suppression of unfair competition in the market would result in the elimination of the gray economy. In addition to their thesis, the Ministry announced that 13,000 fiscal payers applied for their business facilities to move to the new model [49].

Due to the inadequate action of the Ministry regarding the implementation of the mentioned Law, on 14 March 2022, a protest of market vendors was held in Belgrade, organized by the "Opstanak" Association [50]. The Ministry of Finance did not oppose the possibility of the protest as long as it was in accordance with the regulations in force

(without jeopardizing the work of markets) [51]. The protests and the boycott provoked by the Law on Fiscalization spread to other big cities. One of them was Kragujevac, a former big industrial center in Central Serbia. Many people who had been made redundant found their salvation in markets. However, now there is a tendency to "put a padlock" on the stalls since the sales fell, and the situation is very hard. "*In order not to turn Kragujevac into a valley of famine, and to avoid social problems, an appeal was issued to the authorities to review the factual state, which can only be seen in the field, on the faces of these people*" [52]. Closing markets would not do any good to anyone, especially to the country and to the local self-government. A similar state was also seen in the markets of Novi Sad, the capital of the Autonomous Province of Vojvodina (Northern Serbia). The introduction of fiscal cash registers was seen as an "attack" on traditional business performance. "*If they impose cash registers to us, at least 50% will leave the markets immediately. Since there are some who believe they will survive, we are giving them a month or two to realize that they will not survive, and then they will leave as well*" [53].

After the failure to reach the agreement, new protests were held. So, on March 21 and 30, several dozens of merchants gathered in front of the General Secretariat of the Republic of Serbia in Belgrade. An open letter was sent, and a petition was handed over with 15,840 signatures against e-fiscalization. The support came from the registered agricultural households, as well as from those within the art and artisanal activities, which would not be included in the fiscalization. The signers believe that, among other things, "*market vendors will not be able to adjust their prices during the day in order to prevent their products from perishing, or to set their work hours in accordance with extreme weather conditions . . . Non-compliance with the provisions of the Law leads to the implementation of very harsh penal policy*" [54]. However, the authorities of the Tax Administration claimed that they would have to own cash registers so that it could be established who earns how much. Namely, the VAT system would not include entrepreneurs who earn less than 68,000 USD a year [55].

Not satisfied with the response of the state, market vendors from the whole of Serbia blockaded the roads in Belgrade on April 13. One of the organizers said: "*The people of Serbia have been humiliated once again, nobody wants to talk to us, but we will not give up on the struggle for our existence*" [56]. They conclude that the attitude toward the vendors was humiliating, which the state should be "ashamed" of. The representatives of the authorities remained "implacable" despite the growing pressure. They pointed out that everybody who does business in retail sales has to have fiscal receipts. "*No reseller of goods will be exempt. End of story. And that will start on May 1*" [57]. This would make a distinction between individual producers of fruit and vegetables and those who sell consumer goods and food products of other producers.

However, the stance of the authorities was modified after the market workers' meeting with the president of the country. The vendors asked for a special status. As their main argument, they presented the fact that they do not reach even one-half of the annual turnout projected for that model of tax payment. Also, they pointed out that they already pay flat-rate tax, which is 150–200 USD, depending on the local self-government [58]. The state authorities agreed that a "constructive solution to the problem" should be reached. With that aim, a workgroup was formed in order to find a solution regarding the preservation of markets. This primarily refers to the option of tax collection from the sales in the markets, which would lead to the better control of the goods from the viewpoint of the authorities and the improvement in the working conditions from the perspective of the vendors. A consensus was reached that they should strive to find a solution where all the actors could be winners [59].

As a result of the negotiations, during the second half of 2022, in December of the same year, a decision was made to postpone the fiscalization of markets until January 2024 [60]. State authorities explained the inability to create adequate conditions for its implementation due to the political conditions (author's note: snap elections) and the change in the legislation (the Law on Communal Activities). A consensus was reached

that the precondition of all the questions is an improvement in the "work environment" in markets [61].

## 3. Methodological Framework

The research process comprised several segments. The analysis of the market activity in Serbia represented the first phase. Through the overview of the historical and social development of markets, their importance was stressed, both in the past and in modern times. The definition of the status of market activity in the national legislation, as well as the challenges that accompany it, were the basis of the second phase. The focus of the third phase was the analysis of the economic role of markets in the current market environment. A particular segment (the fourth phase) was dedicated to the process of electronic fiscalization as the pioneer venture in this activity. All of the mentioned aspects (phases 1–4) are presented in the first part of the article, in the Sections 1 and 2. For the abovementioned purposes, the relevant literature (printed and online) was consulted, as well as the legislation and data from the state authorities (Statistical Office of the Republic of Serbia, responsible Ministries).

As a novelty, the introduction of fiscal cash registers to markets led to a polemic in the public space since this process incorporates many actors. Generally, three large groups are included. They are the state authorities (author's note: the Ministry of Finance–Tax Administration), tax payers (author's note: vendors), and also the end users (author's note: customers). Each group was interested in the activities in this field for different reasons (motives). "Some say this, some say that" is a phrase that may be the best description of the state of public opinion regarding this question. Therefore, on one side, we have an almost exclusively positive opinion, while on the other hand, there is strong opposition. Nevertheless, a specific group expresses a "more real" (neutral) stand. For all these reasons, in the last (fifth) phase, empirical research was carried out based on a qualitative analysis of the media content regarding the question of the electronic fiscalization of market activity.

According to the data from the Business Registers Agency of the Republic of Serbia [62], there were 2508 types of media registered in the country in 2022. The largest number of them were print media (937/37.4%), internet portals (751/29.9%), 333 radio stations (13.3%), and 244 TV stations (9.7%), as well as 28 news agencies and 84 internet stations which offer media information. Due to the speed of communications in the media space, digital media is of special importance. Among them, the most prominent ones are online versions of print media and numerous internet portals. They belong to one of the five groups known as "hubs" which enable people to access information, education, entertainment, and cultural self-realization using the internet, i.e., the benefits of the Web 2.0 service [63].

Thus, this research involved qualitative content analysis as a method directed towards meanings and interpretations instead of concentrating only on numerical counts or variables-based causal modeling [64]. An analysis of media content (news) and users' (readers') opinions related to the implementation of electronic fiscalization in communal activities, with the emphasis placed on market activity, was performed. It comprised texts from the leading digital issues of print media, as well as the representative internet portals, taking into consideration their geographical distribution as well. Using adequate software (ATLAS.ti, Nvivo, and others), the content analysis proved to be adequate for the examination of comments (opinions, perception) from the readers/citizens [65–67], and has been used in political, social, health-related, and other studies [68–75]. These studies served as a starting point for the application of the methodology used in the previous period in various research areas in several countries.

Among the 23 media outlets (Table 3), the majority have national coverage (79.1%), while others have regional character. In total, their participation is as follows: daily/weekly newspapers (39.1%), television broadcasters (30.4%), web portals (26.1%), and radio stations (4.3%). With the exception of the national broadcaster, Radio Television of Serbia, all other media are privately owned.

**Table 3.** Context of texts dedicated to e-fiscalization in Serbia per type of media (in numbers).

| Source | Location | Type | Section | Context | | |
|---|---|---|---|---|---|---|
| | | | | Positive | Neutral | Negative |
| RTS | Belgrade | TV/portal | Society | 2 | | 1 |
| 021 | Novi Sad | Portal | Business | | 1 | 2 |
| Voice | Novi Sad | Portal | Analytics | | 1 | |
| Nova | Belgrade | TV/portal | Society | | | 1 |
| Euronews | Belgrade | TV/portal | Agribusiness | | 1 | |
| Informer | Belgrade | Newspaper/portal | Politics | 1 | | |
| Večernje Novosti | Belgrade | Newspaper/portal | Economy | 2 | | |
| Kurir | Belgrade | Newspaper/portal | Business | 1 | | |
| Glas Šumadije | Kragujevac | Portal | Society | | | 1 |
| RFE Balkan | Belgrade | Radio/portal | News | | | 3 |
| Danas | Belgrade | Newspaper/portal | Economy | 1 | | 2 |
| Vreme | Belgrade | Weekly/portal | News | | | 1 |
| Sputnik Serbia | Belgrade | TV/portal | News | | | 1 |
| Telegraf | Belgrade | Newspaper/portal | Business | 1 | | 2 |
| N1 | Belgrade | TV/portal | Business | 1 | | 1 |
| Blic | Belgrade | Newspaper/portal | Business | | 1 | |
| Nova ekonomija | Belgrade | Magazine/portal | Economy | 1 | 1 | 1 |
| Politika | Belgrade | Newspaper/portal | Economy | | 1 | 1 |
| B92 | Belgrade | TV/portal | Business | 1 | | |
| Direktno | Belgrade | Portal | Society & Economy | | | 1 |
| Južne vesti | Niš | Portal | Economy | 1 | | |
| Mondo | Belgrade | Portal | News | | | 1 |
| Tanjug | Belgrade | TV/portal | News | 1 | | |

Source: [42,46–48,50–59,61,76–99].

In accordance with the abovementioned, this segment of the research should answer the following research questions:

(1) What is the dominant discourse regarding the fiscalization in the public space?;
(2) What are the main pros and cons of fiscalization?;
(3) What are the reactions of vendors (engaged persons) regarding the effects of fiscalization?;
(4) What are the reactions of customers (users) regarding the effects of fiscalization? and
(5) What are the possible solutions for overcoming the disputes regarding the state–market vendors' relationship?

Respecting the principles of qualitative research, we selected 38 texts and relevant digital media content (Table 3), posted from April 2021 to the end of December 2022. Among the analyzed headlines, nineteen had negative (50.0%), thirteen (34.2%) had positive, and six (15.8%) had neutral content. They belonged to different thematic sections, mainly business and economy (52.2%), but also news (21.7%), society (17.4%), and politics and analytics (8.7%). At the same time, 268 out of 315 (85.1%) comments on the listed news were included in the analysis. The rest of the comments were eliminated due to inappropriate content. Taking into account the comments included in the final analysis, 115 (42.9%) had a positive, 87 (32.5%) a negative, and 66 (24.6%) had a neutral attitude towards the issue of fiscalization. It should be noted that all the content, because of the formulation of the research, was in the Serbian language.

For qualitative analysis of data, the widely accepted open-access LiGRE software (https://ligresoftware.com/ (accessed on 15 July 2023)) was used [100]. First, after selecting media content and collecting data, the lists of text content and the comments were made. The data were then imported into the software. For their easier interpretation, the coding of comments (responses) was performed according to categories using the mentioned software tool. In the last stage, an analysis was performed that enabled the generation of results. This made it possible to evaluate the relationship between different data and conduct searches to support our conclusions. It is worth noting that, in relation to the access

used in the qualitative analysis defined by Patton [101], and in the research performed by Gao and Koo [68], induced access was used here as well. The reason for this is the absence of a determined methodological framework which defines the factors that will be included or excluded.

## 4. Results

The research we conducted provided a great deal of information related to the implementation of fiscalization in Serbian markets. In addition to learning about the historical, social, and economic significance of markets through time, our detailed analysis draws attention to the contemporary challenges of incorporating modern business practices into markets. One of the most prominent innovations is the introduction of fiscal cash registers. Therefore, an analysis of available readers' comments on online media content related to fiscalization policy in Serbia is presented below.

*Public Discourse: The Analysis of the Media Content*

In the first phase of analyzing the media contents, using the LIGRE software, 56 codes were singled out to address the research questions. In the next step, the number was additionally reduced, and four main codes ("vendors", "customers", "state authorities", and "professional associations") and nineteen additional codes (Figure 1) were selected from the digital content ("online newspapers/magazines" and "websites/portals"). At the top of the list, there were those media which became the creators of public opinion due to their role as the "distributor" of content (news). With their information on e-fiscalization, they united all of the opinions/stances "for and/or against" that were mentioned by the actors included in the abovementioned process. Based on the evaluation of texts and the follow-up contents (author's note: readers' comments), several types of discourse were noticed. Dominantly present were the ones regarding economic, political, and social aspects, but there were also notions of others (e.g. technological aspects). Taking into consideration the complexity and scope of e-fiscalization, there was a harsh polarization (distinction) in terms of the expressed positive, negative, and neutral stances regarding the given question.

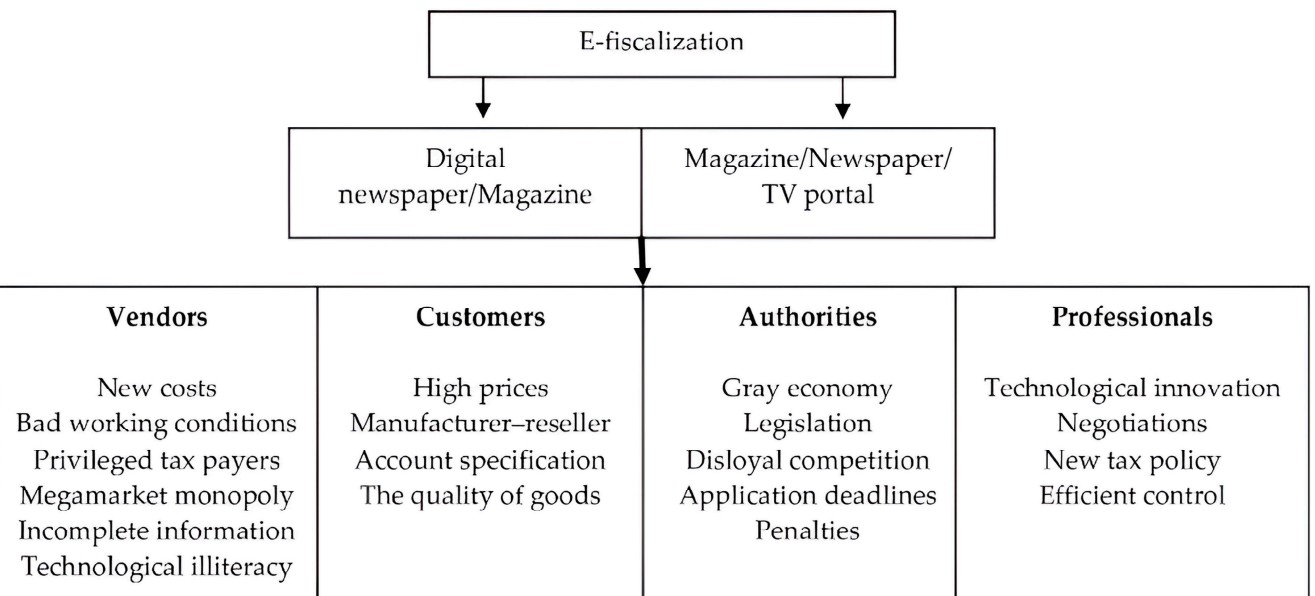

**Figure 1.** Result of iterations of open coding. Source: Authors' elaboration.

The analysis of the media content showed a strong ***affirmative reaction*** from a part of the public (author's note: customers and VAT payers) regarding the e-fiscalization of market activities. Some of the opinions were "they should not differ from the others". From the economic point of view, a large number of participants believe that markets

should be fiscalized "unconditionally" because "*95% of so-called producers and vendors of agricultural products are actually resellers*". The state must not "miss" the opportunity to establish order in this market. In that case, it would be a "*direct strike directed toward an agricultural producer who deals with production and sales risks*" (Antonije) [54]. The prevailing opinion was that there should be no privileged categories. Tax collection from all the citizens, including market vendors, lawyers, taxi drivers, etc., enables the collection of assets for pensions and other social welfare expenditures, as well as the construction of infrastructure. Why would all the shop owners not dispose of their cash registers then? If we want a regulated, democratic, and stable state, then there must not be any gray economy. An average customer wants to see what they have bought on their receipt and where that money goes. "*When I buy ice cream, newspaper, cigarettes, I get the receipt at every booth. However, if I spend a much larger amount of money on the market, there is no receipt*", is a statement from Mojica from Subotica [76].

Certain participants/readers (comment writers) point to the positive examples from other European countries. So, Jelena, who lives in Switzerland, went a step further and said that "*cash registers are mandatory on the markets in West Europe, and there is also a possibility of paying by cards*" [55]. She considers it a standard way of doing business. A similar comment was left by Vlada, who says that vendors in Slovenia, Poland, and the Czech Republic have cash registers and POS terminals for cards and wonders why Serbia is so far behind [77].

In the media space, there are also opinions directed straight to the vendors who oppose the introduction of cash registers. Particularly prominent is the opinion of Marko from Belgrade, the former worker in sales who says: "*Perfume shops issue receipts, which is 20%, and then they pay the income of about 10%, the tax on employees' salaries, etc... Market vendors with no receipts and slightly lower prices, and often with the same ones, are the unfair competition with their daily profit of at least 215–322 USD*" [57]. Regarding the tax (VAT), a harsh comment was left by Milan who said that the government should be more persistent and more rigorous. "*That employee does not pay any social, pension, or healthcare taxes, so when they get really old, they will blame the state*" [76]. He also asks where the money in the public budget will come from, how the roads will be built, or kindergartens, or hospitals. A similar opinion is shared by Saša who blames the country for this "confused" state. "*The state embarrasses itself by bending before smugglers and resellers. So, all of them who pay VAT and have cash registers should blockade the government because of the unfair competition*" [78].

As one of the mentioned reasons for the fiscalization of the market sector is the ability of guaranteeing quality and quantity of goods, it was also stated that markets are not what they used to be either. "*You always knew everyone, everybody had their vendor, you knew which village they came from, whether they treat fruit and vegetables, and whether they respect the waiting period*" [48]. The e-fiscalisation of markets has removed the social fabric of markets by dehumanizing interactions that have traditionally been at the heart of market culture. Additionally, customers also point to the increasingly "worse quality of goods" (rotten cherries, overripe strawberries) that the vendors "quickly put into bags" and for which there is no possibility of refund. "*On the markets, we buy a pig in a poke, or it looks like Russian roulettewhen it comes to the expiry date and pesticides, fungi, parasites, etc*!" That is why some sort of control is essential for the protection of customers. With a receipt, a vendor provides a guarantee for the goods. "*To ask for and to get a receipt is a legitimate thing!*"—these are the words of a retired vendor from Novi Pazar [79]. They also believe that a small percentage of vendors (resellers) show respect for customers. "*On the market, they set the scales the way they like, and when you weigh it, there is always around 300 g of fruit or vegetable less*", says Dušan from Kragujevac [51]. The ability to choose the goods and services is the main motivation for going to supermarkets.

Also, the conditions of blockades of roads and impeded everyday functioning cause additional opposition from customers toward market vendors. This can be best noticed in the comment by Goran, who says that he will boycott the market because they blocked the city and caused problems. "*I will buy in supermarkets, even at higher prices*" [58]. The same opinion was shared by Zorica and Milan from Belgrade, who saw those protests as

disruptive for citizens. "*As we can see, if somebody does not like something, or if they believe they have suffered some kind of injustice, the easiest way to show their dissatisfaction is to blockade the streets, and then everybody suffers the consequences. Besides, we believe that there are much more resellers in the protests and much fewer peasants*" [80]. The fact that they are not satisfied by the state in the country and in society is not the reason for irresponsible behavior. "*There were opportunities for changes, but they kept silent. In the previous elections, almost 70% voted for cash registers. Now they have nothing to oppose to*" are the words that can often be heard in public [81].

Of course, we must not forget the not so small group of existing tax payers who have been operating in hard economic conditions for years. They say that they feel "tricked" by the state. "*You trampled about 200,000 of us who regularly pay all the obligations through various taxes, eco-taxes, and you name it. . . and in the end, we believed in the state that will protect us from unfair competition by new fiscalization*" [78]. Having realized that "disobeying" the law pays off, there are often comments such as: "*We, the entrepreneurs, do not want to use new cash registers either! I am publicly calling for a boycott and disobedience!*" [57]. The "unfair" way of collecting taxes is what Dragan (Kragujevac) talks about. After 23 years of work experience, he had to unregister his shop, and, as he said, he would never open it again. The reason he mentions is the inability to work in the system, which is based on the principle that "*the rule is that there are no rules*" [52].

Quite the opposite, and contrary to the state, which insists on introducing the new tax system in market activity, vendors show a ***negative*** stance toward the "good" intentions of fiscalization. Their opposition was directed at several targets. The very announcement of fiscalization met "distrust" due to the lack of information. It can be best illustrated by the comments from the vendors at the markets in Novi Sad. "*It is still only whispered about among the vendors on the market. They say that we will have to have fiscal cash registers from next year, while some say that it does not apply to us, who sell only our goods. I don't know. We bring products from our household, we make them as a family, and I am usually the one standing at the stall*" [82]. A greater concern is related to the resellers. According to them, they already have low incomes, so they fear that the introduction of VAT will force them to raise the prices of imported fruit and vegetables. That means that they will have to close their stalls because there will be no customers.

On the other hand, many vendors point to the specificity of the work of markets. The lack of knowledge about the work on markets is what "irritates" Srđan, a former market vendor from Prijepolje. Vendors should not be called "smugglers". All the market stalls are bid for, but it is necessary to have a registered activity. They often pay "unreal" monthly rent, as well as daily rent, while the inspections "terrorize" them more than wholesale vendors. Also, they pay income taxes and other expenses, and at the same time, they do not have the assortment of goods to have a turnover like shops do. He finishes with the following words: "*Just because they are small and they cannot defend themselves, they are everybody's target*" [83].

At the same time, Tamara, a vendor in one of the markets of Novi Sad, believes that the opposition to fiscalization is totally justified. The work conditions, high monthly rents for the stall (around 456 USD), for the warehouse (356 a year USD), and the increase in prices of products does not leave much space for earnings (only 0.30 USD per kg of fruit) [54]. At the same time, big shops can send the bad goods back, they have rebates, and take goods with deferred payment, which is not the case in the markets. Her colleague from Belgrade supports this opinion. "*Pay 2.5 USD for the stall every day, bid the stall for 365–820 USD, and every month give the manager 2–3 kg of your goods. On top of all that, there are fiscal cash registers. Who is mad here?*" [84]. First, proper working conditions should be provided to all the markets. In villages, vendors do not even have covered stalls, water, lights, or toilets. "*When it rains, you do not work, when the wind blows, you do not work. Nothing is solved, and they introduce measures*" [85]. This is what Mića from around Leskovac says.

So, besides inadequate working conditions, the costs of introducing and maintaining the system are recognized as the biggest problem. "*In the shops where the value of goods is*

*21,500 USD there is one cash register, and I need to have two cash registers for my two stalls. Also, the value of my goods does not exceed 1650–2150 USD. Even if I sell all my goods, I cannot cover all the costs required by the state*" [86]. It is essential to be aware of how things work in the markets, and it is evident that the lawmakers did not go out into the field. If everything is to be put in order, then we should all be equal, and it often happens that the "privileged" do not even pay the rent for their stalls. That is what an elderly vendor on the market in Sremska Mitrovica thinks.

Even though most of the customers express their support for the process of fiscalization, some of them show understanding and support for market vendors and their cause. The reasons are different. On the one hand, there is a risk of a significant rise in the prices of the goods which are still quite affordable. This attitude is also shared by Marko from Niš who believes that "*if they are fiscalized, the prices will rise by at least 20%, and the earnings will fall*" [61]. According to him, the products in markets are "already expensive enough", and with the fiscalization, it would be pointless to go shopping. However, there is the awareness that giving up on this process would mean returning markets to the streets, as it was in the 1990s when the gray market "flourished". Also, as Ana points out, at the market, "*you can bargain, which is not the case in other forms of trade*" [87]. This opinion is also expressed by Goran and Verica, who see the advantage in the possibility of changing prices during the day, depending on weather conditions, number of visitors, and the level of goods freshness. They believe that the price on green markets must be determined as on a stock market exchange. "*It is necessary that it changes during the day because the goods lose their weight. Also, a lot of goods are thrown away, and nothing can be earned*" [88].

Also, some citizens show their support to the vendors through criticism of the state activities in the field of fiscalization. Mimi ironically says that the state wants to acquire the assets everywhere and adds: "*They are so hungry for money that they will start taxing even the hole on a flowerpot*" [89]. A similar attitude is expressed by Jelena, who defends the "getting rich" of market vendors by saying: "*Come, you, gentlemen, there are free stalls on every market and work their every day non-stop at the temperatures for −10 to +40*" [90]. Jovan also points to the risk that supermarkets will eventually rent market stalls. He expresses his opinion interestingly by saying "*Do not dare to allow ordinary people to earn some money. Try to take their souls. They hardly earn for bread, and cash registers are the only thing they need now! Farmers are leaving their health in the field, and they should be offered benefits, not new expenses*", a sentiment shared by Milica [91]. Special risk is perceived in "promoting" big retail chains compared to small, ordinary vendors. As Rade from Vladimirevci says, farmers are especially endangered in the biggest cities where there are retail chains. If they keep insisting on fiscal cash registers, nobody will find it worthwhile to work on markets. "*They can comfortably put padlocks on their doors*" [88].

However, certain participants point out the difference between producers and resellers. It often happens that vendors go to supermarkets and buy everything on sale, and they resell the same goods on markets at extremely high prices. This phenomenon is becoming very frequent because there are no limits on buying such goods in supermarkets (not referring to the essential provisions). Due to the abovementioned, a large number of agricultural producers say that "*the whole world helps the primary agricultural production, and our country treats its citizens like an evil step-mother*" [77]. At the same time, a comparison of other vendors with those on the markets is not fair, as Jasmina from Zaječar believes. "*You are 'noblemen' for them. You work inside, while they are outside all the time. They earn 30 times less and do not have the basic conditions for work*" [92].

There is also a psychological fear related to the "social" role of markets. They say that cash registers do not guarantee the quality of products. They believe that "bargaining" and conversation with the vendors are absolute pleasures. Namely, "*people love to look around, talk, and taste products, and if cash registers are introduced, that spontaneity will disappear from the markets forever*" [93].

It often happens that "the truth is somewhere in the middle". Among the comments, there are quite a lot of them with a ***neutral (indecisive) attitude*** toward fiscal policy. A

lot of citizens believe that a certain kind of "financial" control should be introduced at the markets, but that fiscal cash registers would be a problem. They primarily have in mind the "technical illiteracy of the users" and their maintenance. Naturally, the prices must at least be clearly displayed, and some sort of block-receipts could be used for a specified kind and quantity of goods. Finally, you can introduce a flat-rate tax for market sales (Nikola). Also, Miloš shows a certain level of understanding for market vendors. Exception from fiscalization is not a permanent solution, but why attack market vendors? "*Except a part of tin, they do not have anything–no electricity, no water, but only the open sky above themselves, and they are already paying the tax and stall rents too much. Most people may not be aware of that, and they attack people from the lowest social layer*" [82,94].

A need is also noticed here for a distinction between vendors on one side and resellers on the other. One of the interviewees suggested a solution where cash registers should not be introduced for producers when compared to other vendors. "*A lot of money is turned here, and they pay their employees very low salaries, sometimes below the mandatory minimal amount. And they are often not even registered as employees*" [90]. This would make a real distinction, and the consumers would not be deceived (Dejan). Ljubomir from Varvarin agrees with this opinion. "*You don't pay much when you buy from peasants. They are the only ones who are not charged for their hard work. From seeding to harvesting, the process is very long and hard. Resellers's earnings are 100% (and even more), and they blackmail peasants all the time. They are the ones who should be obliged to have fiscal cash registers*" [57,84].

However, Milica also left an "edited" comment that refers to the introduction of fiscal cash registers. She believes that a receipt should be issued for each item, as everywhere in the world, but the producers should also be protected. Only if they can charge their work properly, can they be in a better position than resellers. Darko from near Belgrade offers a "Solomonic judgment" [95]. Tax policy and origin control of the goods should be defined in order to protect both vendors and resellers. Leona goes a step further and suggests an innovative system of "closed market" type—you buy what you want, from whom you want, and pay at the exit (in cash or by card).

Zdravka, a watermelon producer, has an interesting way of thinking and says that producers and resellers are well-connected. Namely, "*a peasant cannot be present at the market because their products come from the fields every day*" [96]. They would need dozens of workers and stalls to "place" their goods on markets, which is not possible. In that way, a "peasant's" balance would be negative, and supermarkets would regain their monopoly. In relation to the previous comment, there are (although rarely) thoughts such as the one presented by Robert from Sombor. He says that producers very often sell products that are not theirs on their stalls in order to enrich their assortment and make it more attractive. Thus, it would be hard to prove which goods they did not produce.

Most of the actors agree that unfair competition is an equal danger for all the vendors in the markets. As suggested by the owners of registered agricultural households, the attitude of the authorities toward them is killing their motivation for doing business. Namely, they collaborate with the competition that sells the same/similar products through various types of pressure (e.g., frequent inspection surveillance). In their opinion, the major problem is the existence of so-called "smugglers". "*They smuggle fruit and vegetables from all around, without quality check, and they put customers at risk*" [46,97]. Naturally, in everyday business, many organizational problems occur. In that respect, a difference should be made between big and small places. In most of the settlements in Serbia, markets operate on certain days (most often twice a week). Thus, vendors have to go to other places, which augments the costs and affects the quality of goods. This causes the problem of price indexing and receipt issuing due to various administrative regulations of self-governing units.

In general, almost all the participants highlight that "imprudent and unplanned" political solutions to economic problems often cause additional trouble. In that sense, a certain number of citizens believe that a policy maker has not taken into consideration all the aspects of this question. "*Market stall cannot be regarded the same as enclosed space, whatever is sold on that stall*" [98]. Also, there are questions about the labor market since the

majority of older vendors could not find another job. Thus, a lot of them believe that the authorities are entering a very risky business with these actions and that it will be too late when they become aware of them. "*Whoever started establishing an order in Serbian markets, left politics very quickly. The roles there change rapidly*" [42].

## 5. Discussion and Conclusions

This study provides an evolutive contribution to the attempts to put local markets into the focus of research. Although it has a multifunctional significance, the studies related to various aspects of market activities in the wider area of South-East Europe have not been represented enough. Also, taking into account the characteristics of former socialist transitional societies (including Serbia), this question is becoming even more important. In the previous period, initial studies [6,9,22,102,103] started to emerge in this field. This research pointed to various aspects of the "specificities" of market activity. On the one hand, this form of trade has always been available to all social and educational groups and enabled the socialization of the participants in the process.

On the other hand, in economic terms, market activity has a significant potential. Namely, the total turnover of goods on markets is rising, bearing in mind that a significant share of customers have a positive attitude (loyal relationship) related to this type of shopping. A market possesses the capacity to adapt to various socio-political and economic challenges, which was especially notable in the 1990s (international sanctions, inflation, gray market, etc.) [35]. Therefore, it can be said that it is an important link within the overall economic development. In modern circumstances, adjusting to business trends is essential. However, in Serbia, this sector has long been out of the "main streams". The greatest weaknesses are reflected in poor infrastructural equipment and inadequate management (absence of vision). The problem became more complex with the entrance of fair/unfair competition (supermarkets) and with the strengthening of other types of trade (online commerce), with more frequent contactless payments (use of credit cards).

In the current unenviable conditions, each novelty brings uncertainty. Among the numerous ones suggested, the electronic fiscalization plan for communal activities has been marked as a big step in doing business so far. However, even its announcement and then the adoption of the Law on Fiscalization faced divided opinions from the public. Protests accompanied harsh polarization during the negotiation process which lasted for many months. At the same time, the image in the media related to that question greatly impacted the shaping of public opinion. State authorities pointed to the positive effects of pushing market activity into the new economic framework after so many years. This would ensure the collection of additional financial assets, but it would also control business subjects more effectively. Also, of no less importance is the suppression of the widely spread gray economy.

On the contrary, market vendors and professional associations pointed to the negative repercussions of the implementation of the Law. They saw this legal suggestion as an "attack" and an ill-intentioned (unplanned) move. The opposition to the process of fiscalization was accompanied by mass protests. At the same time, as one of the interested parties, customers expressed their suspicion about the "innovated" system in the sector of market activity. Namely, while in the public discourse the policy makers only perceive the benefits of tax introduction, and the vendors see new costs, the customers do not have a clearly defined view (indecisive attitude). The analysis of the media content points both to the opposition to (negative attitude) and to the support (positive attitude) for electronic fiscalization. The reasons for both attitudes are numerous, but they all agree that it is necessary to start from the "cause rather than from the consequences".

According to the comments of digital content users, it is possible to single out several theses that support the introduction of fiscal cash registers. First, a fair position for market participants would be established. The existence of double standards negatively affects the entire tax collection system. It must be taken into account that the funds from the collection of this as well as other taxes serve to finance numerous sectors in the country (health,

education, social protection, infrastructure construction, etc.). Also, an organized system means safe product quality and more efficient control, which was called into question in the previous period because marketplace activity was exempt. Basically, the system has to be established, or it would collapse like a "tower of cards".

The opinion of the opponents to the introduction of this type of fiscalization in the marketplaces was undoubtedly shaped to by the lack of information and public discussion among all stakeholders. Actually, market vendors pay a number of obligatory charges (rent, storage, etc.) to the local authorities (although they are not liable for VAT). Any additional levy could cause more problems, both for the merchants and for the end users (customers). On the one hand, the new costs will drive a significant number of vendors to the street, which will return them to the "gray market". On the contrary, it will lead to an unprecedented rise in retail prices and an additional decrease in the purchasing power of residents. Resistance to fiscalization is additionally conditioned by the fact that other sectors are exempted (lawyering, taxi transport, religious activities, etc.).

Specific fiscal measures must be implemented, but numerous specificities must be taken into account (primary producers vs. resellers, small vs. large vendors, working conditions and storage of goods, etc.). There must be communication at all levels. In essence, all actors are part of the system but have different roles and positions. It is necessary to respect all the differences within and the diversity of this sector. In accordance with the above, it is possible to apply different tax incentives and other types of utility benefits (rent, water supply, sewage, electricity, etc.) to specific categories of merchants.

It should be noted that the Serbian market sector has come through very challenging political, economic, and social changes [6]. All the actors on the market agree that modernization of all the amenities related to markets is necessary, but the order of activities is "disputable". Therefore, in order to overcome numerous obstacles in the future, some measures must be implemented. Firstly, a favorable environment must be created for the inclusion of small entrepreneurs as an essential link to local economies [102,104]. Also, it is necessary to stop the rapid negative trends. This primarily refers to the change in the age structure of vendors, which would make local markets more attractive for "modern" customers as well. Secondly, besides the quality of the product, they should also have competitive prices compared to retail chains.

Last but not least, there is the question of the technological modernization of the markets. It is necessary to improve the existing (multi-decade-long) model, which would ensure their survival in the future. If this activity did not adjust to the business model of the 21st century, it would be "the last nail in the coffin" of markets. Therefore, due to the complexity of this question, besides the representative associations, other authorities should also be included, especially the Ministry of Construction, Transport, and Infrastructure (responsible for communal activities), as well as the Ministry of Agriculture, Forestry, and Water Management and the Ministry of Trade. Further investments into infrastructure, incentives for households and entrepreneurs, and strengthening capacities (innovativeness) will enable the inclusion of market activity in Serbia's new agricultural policy.

*Limitations of the Study and Future Research*

Of course, as with most of research, this one also has its limitations. On the one hand, there are technical difficulties related to the selection of representative textual content. Also, their correct interpretation using adequate software is a difficult task. While the application of qualitative methods has specific challenges, the approach has its advantages in certain segments (analysis of subjective experiences, opinions/attitudes, perception through observation, interviews, etc.), which enables a comprehensive analysis. However, the use of standardized instruments and quantitative data (e.g., data obtained in a survey) provides scientific objectivity [105–107]. The application of combined methods (qualitative and quantitative) is convenient for testing the existing theories (and hypotheses), in which available (free) and reliable software enables an overview of the broader picture of the studied problem. Therefore, future studies should be carried out in more directions. First,

by using interviews and survey research as research techniques, various actors should be included in this process (customers, vendors, members of professional associations, representatives of the national and local administration, etc.). Such an approach would provide additional and more complex answers to the research questions related to the functioning and significance of the market activity in Serbia. The findings derived from this and other research can serve as a starting point for formulating future strategies and policies at all levels (national, regional, and local). The cooperation of all stakeholders would be in the public's interest in comparison to the individual requests that took priority in the previous period. Furthermore, since the qualitative analysis of the media content comprised the media in Serbia, similar actions could be carried out first in the countries of the former Yugoslavia and then in other so-called post-socialist countries in East Europe and Asia. This will enable a comparative analysis, highlighting the same/similar problems, as well as possible solutions. The results of research from geographical regions of similar historical, political, and socio-economic development would serve as a good basis for acquiring new knowledge and experiences and as a guideline for defining future policy in this field.

**Author Contributions:** Conceptualization, S.D. and M.D.P.; methodology, S.D.; software, S.D.; validation, M.D.P. and Z.V.-M.; formal analysis, M.M.R.; investigation, S.D. and M.D.P.; resources, Z.V.-M.; data curation, M.D.P.; writing—original draft preparation, S.D.; writing—review and editing, M.D.P., Z.V.-M. and E.E.-L.; visualization, E.E.-L.; supervision, M.M.R.; project administration, E.E.-L.; funding acquisition, S.D. All authors have read and agreed to the published version of the manuscript.

**Funding:** This study was supported by the Ministry of Science, Technological Development and Innovation, Republic of Serbia (Contract No. 451-03-66/2024-03/200172).

**Institutional Review Board Statement:** Not applicable.

**Informed Consent Statement:** Not applicable bearing in mind the specificity of this study. Namely, these are publicly available statements of respondents (media content), so obtaining consent is not necessary.

**Data Availability Statement:** Data are contained within the article.

**Conflicts of Interest:** The authors declare no conflicts of interest.

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
