# Peer review of "What Are the Current Directions in the Local Marketplaces Fiscalization? The Online Media Content Analysis"

_societies, doi:10.3390/soc14040053_

Round 1

Reviewer 1 Report

Comments and Suggestions for Authors

The introduction of the article presents on commonly known questions about the nature of the market and types of markets. The focus should include the importance and actuality of the research. Why fiscalization of local marketplaces is important? Why should stakeholders' opinions be explored on the issue of fiscalization of local markets? The aim of the research is not clearly described. Table 1 is not clear. Check data. Is it correct that turnover of agriculture products in Serbia for a year (2022) was a half million USD? Also see line 189 (turnover 23.2 million USD). Different currencies were used in the text (USD, EUR, RSD, dinars). Please use only one. There was not information about the total number of studied publications? What are the criteria for analyzing only 39 texts and 268 comments of them? Do statistical analysis of the posts and the texts? For example, share of print/electronic media, share of positive/neutral/negative opinions, share of national/local media. I think 4.1. Introducing Regulations for Electronic Fiscalization of Market Activities should be in Section 2 Background of Markets in Serbia. All opinions in 4.2. Public Discourse: The Analysis of the Media Contents should be referenced. Add a section in article with scholars' results (on the issue of fiscalization in other countries/similar issues of public importance) to support the approach that you used. The findings are too general. It is not clear who would benefit from them and how. It is not clear what problem they are solving. Please rewrite this part and be more specific.

Author Response

We are sincerely grateful for the useful comments on our paper. In the text, all corrections were made by using Track Changes option.

Please see the attachment with the responses to your comments. 

 We hope that the revised version of the manuscript will better meet the journal's requirements.

Best regards

Reviewer 2 Report

Comments and Suggestions for Authors

First, I appreciate the opportunity to be a reviewer for such an article. The authors of this article attempt to present new proposals for measures and public reactions against the electronic visualization of market activities, which the majority of Serbian society finds unprecedented. Research concentrates on the historical evolution, electronic visualization of market activities, and markets' economic importance. The aim is to determine the differences and parallels between markets' social and economic perspectives using theoretical assumptions and qualitative content analysis at a critical historical moment for local markets in Serbia.

 After reading the research and analysis of the authors of the article, I would make the following comments and suggestions:

I. Originality:

The research topic of the article is relevant because markets have an economic role in a country's financial processes. In addition to their economic role, they create an opportunity for a social environment, which is extremely important for recovery after the Covid-19 pandemic. At the same time, the need for the fiscalization of economic processes has always been a matter of discussion, and even more so - how the revenue state administrations should regulate this process. For these reasons, I admire the authors for deciding to explore this issue.

II. Literature review:

The used literary sources are 93, of which 29 articles, 16 normative acts (primarily Serbian) and 48 electronic sources (mainly in Serbian), which to some extent proves the diversity of ideas with which the authors got acquainted and based on which they expanded your field of knowledge. Eight articles are from the last five years (2019 - 2023), which does not add to the relevance. The authors should make further efforts to this part, despite their claim that such studies in theory and practice are lacking.

III. Methodology:

The Methodology used is clearly described but is based on analyses of other studies and opinions. There are no static methods to show any relationship between the investigated components. For example, based on other studies on the shadow economy and public perceptions of cash payments, the authors carry out two of the 5 phases. It needs to be clarified whether the studies on which the authors rely cover the necessary critical mass of respondents. What are the questions? How are the individual results researched? An analysis of other analyses cannot give a precise result.

IV. Results:

In the Methodology, the authors put 5 phases in which the research took place. The first phase is developed in the first part of the article. For the implementation of the second and third phases, the authors have decided to present it in the "Results" section. However, it is necessary to specify the individual results with specific data and more author conclusions. Thus, it needs to be clarified; the reader gets confused about where these results come from and why they are indicated.

V. Discussion and Conclusion:

The discussion and conclusion are well-formed and based on the data analyzed in the results and the Methodology used. Based on this, the authors make generalizations and conclusions, which prove their opinion regarding the studied issue.

V. Quality of communication:

The article is well-written and easy to read. The authors have considered an issue that is current and debatable. However, there is a slight discrepancy between the title and the article's content, more specifically, in the part that examines the media's attitude towards visualization. I think it would be easier for authors to change the title than the content. Furthermore, it is necessary to expand the Methodology used. Without detracting from the efforts of the authors and the not small work they have managed to do, I believe that by making the extra effort, they would have contributed to the improvement of the article and that it would be one of the contributions to change the economic environment of the local markets in Serbia.

Author Response

(The authors gave the same response as above.)

Reviewer 3 Report

Comments and Suggestions for Authors

I find this paper very well organized and it has implications for practice. However, I believe that the novelty of the study should be better outlined and the authors should discuss more about the broader implications of this study as it concerns only a specific region.

Author Response

(The authors gave the same response as above.)

Round 2

Reviewer 1 Report

Comments and Suggestions for Authors

The authors have taken into account my notes and comments. 

NB Recheck the References. There is two lists. The second one (after No 108) should be deleted. 

Author Response

Dear reviewer,

Thank you for your observation. You are absolutely right. There is an additional list after reference No. 108. However, after accepting all changes (Track Changes option), the list will be eliminated, leaving a single (final) list of used references.

Also, we thank you once again for the useful comments that improved the quality of the manuscript.

Best regards

Reviewer 2 Report

Comments and Suggestions for Authors

First, thank the authors for taking the notes seriously and removing many. Research questions and hypotheses are still missing. If somewhere in the text there are them (I read very carefully and on "search", entering "hypotheses" or "research questions," the results I am looking for do not come out.

I still think that the basis on which the analysis and opinions are made and their focus on the media are incorrect.

I still think that literary sources are not up-to-date enough, and authors turn their attention to Internet resources, such as media sites and electronic journals.

I am finding that Figure 1 is not placed correctly. Also, the numbering of the references used needs to be clarified.

Author Response

Dear reviewer,

We are sincerely grateful for the useful comments on our paper. All corrections were made in the text using the Track Changes option. Minor changes in the second round of reviews are highlighted in yellow.

Please see the attachment with the responses to your comments. 

We hope that the revised version of the manuscript will better meet the journal's requirements.

Best regards

Round 3

Reviewer 2 Report

Comments and Suggestions for Authors

The authors have addressed all of my comments to my satisfaction. I have no further comments/suggestions.